# GeoTMI:
# Predicting Quantum Chemical Property with Easy-to-Obtain Geometry via Positional Denoising

**Hyeonsu Kim**[*]
Department of Chemistry
KAIST
Daejeon, South Korea

**Jeheon Woo**[*]
Department of Chemistry
KAIST
Daejeon, South Korea

**Seonghwan Kim**[*]
Department of Chemistry
KAIST
Daejeon, South Korea

**Seokhyun Moon**[*]
Department of Chemistry
KAIST
Daejeon, South Korea

**Jun Hyeong Kim**[*]
Department of Chemistry
KAIST
Daejeon, South Korea

**Woo Youn Kim**[†]
Department of Chemistry
KAIST
Daejeon, South Korea

## Abstract

As quantum chemical properties have a dependence on their geometries, graph neural networks (GNNs) using 3D geometric information have achieved high prediction accuracy in many tasks. However, they often require 3D geometries obtained from high-level quantum mechanical calculations, which are practically infeasible, limiting their applicability to real-world problems. To tackle this, we propose a new training framework, GeoTMI, that employs denoising process to predict properties accurately using easy-to-obtain geometries (corrupted versions of correct geometries, such as those obtained from low-level calculations). Our starting point was the idea that the correct geometry is the best description of the target property. Hence, to incorporate information of the correct, GeoTMI aims to maximize mutual information between three variables: the correct and the corrupted geometries and the property. GeoTMI also explicitly updates the corrupted input to approach the correct geometry as it passes through the GNN layers, contributing to more effective denoising. We investigated the performance of the proposed method using 3D GNNs for three prediction tasks: molecular properties, a chemical reaction property, and relaxed energy in a heterogeneous catalytic system. Our results showed consistent improvements in accuracy across various tasks, demonstrating the effectiveness and robustness of GeoTMI.

## 1 Introduction

Neural networks have been actively applied to various fields of molecular and quantum chemistry [1, 2, 3, 4]. Several input representations, such as the SMILES string and graph-based representations, are employed to predict quantum chemical properties [5, 6]. In particular, graph neural networks (GNNs), which operate on molecular graphs by updating the representation of each atom via message-passing based on chemical bonds, have achieved great success in many molecular property prediction tasks [7, 8, 9, 10, 11, 12].

However, as many quantum chemical properties depend on molecular geometries, typical GNNs without 3D geometric information have limitations in their accuracy. In this respect, GNNs utilizing

---

[*]Equal contributors.
[†]Corresponding author: wooyoun@kaist.ac.kr

37th Conference on Neural Information Processing Systems (NeurIPS 2023).

3D information have recently achieved state-of-the-art accuracy [7, 13, 14, 15, 16, 17, 18]. Despite of their impressive accuracy, the usage of the 3D input geometry is often infeasible in real-world applications, limiting the 3D GNNs' applicability [19, 20, 21, 22, 23]. Therefore, it is natural to train machine learning models to make predictions with relatively easy-to-obtain geometries. Several studies have investigated the use of easy-to-obtain geometry as input, and it has been empirically confirmed that such geometry can be leveraged to accurately predict target properties [19, 22, 23]. Yet, theoretical basis for fully exploiting such easy-to-obtain geometries to predict accurate target properties remains to be established.

This study proposes a novel training framework, namely "**Geo**metric denoising for **T**hree-term **M**utual **I**nformation maximization (GeoTMI)", which employs a denoising process to accurately predict quantum chemical properties using easy-to-obtain geometries. Throughout this paper, we denote the correct geometry as $X$, the easy-to-obtain geometry (regarded as the corrupted version of $X$) as $\tilde{X}$, and the target property as $Y$. Various previous studies have been conducted on denoising approaches, such as a denoising autoencoder (DAE) [24, 25, 26, 27, 28, 29]. When it comes to predicting quantum chemical properties, the predominant focus of denoising techniques has been on improving the prediction accuracy starting from $X$. GeoTMI, however, aims at improving the prediction accuracy starting from $\tilde{X}$. GeoTMI also explicitly updates the input geometry of $\tilde{X}$ to approach $X$ as it passes through the GNN layers, thereby contributing to more effective denoising. Furthermore, GeoTMI incorporates an auxiliary objective that predicts $Y$ from $X$, allowing it to capture the task-relevant information and ultimately maximize the mutual information (MI) between the three terms of $X$, $\tilde{X}$, and $Y$. The theoretical derivations in this study provide further support for this approach.

GeoTMI offers the advantage of being model-agnostic and easy to integrate into existing GNN architectures. Thus, in this study, we aimed to validate the effectiveness of GeoTMI on different GNN architectures and for various prediction tasks (the nine other molecular properties of the QM9 [30], a chemical reaction property of Grambow's dataset [31], and relaxed energy in a heterogeneous catalytic system of the Open Catalyst 2020 (OC20) dataset [23]). We evaluated the performance of GeoTMI by comparing it to baselines trained only with $\tilde{X}$ and $Y$ using multiple 3D GNNs. GeoTMI showed consistent accuracy improvements for all the target properties tested. In particular, in our experiment on the IS2RE task of the OC20, GeoTMI achieved greater performance improvements than another denoising method, Noise Nodes [27], demonstrating the superiority of GeoTMI. Overall, our findings demonstrate that GeoTMI can make accurate and robust predictions with easy-to-obtain geometries. Code is available on Github.

## 2 Related Works

### 2.1 Predicting high-level properties from easy-to-obtain geometry

Recently, several deep learning approaches have aimed to predict high-level properties from an easy-to-obtain geometry for accurate yet fast predictions in real-world applications. For instance, Molecule3D benchmark [22] aims to improve the applicability of existing 3D models by developing machine learning models that predict 3D geometry. These models predicted 3D geometry using 2D graph information that can be easily obtained and were evaluated using ETKDG [32] from RDKit [33] as a baseline.

There have been attempts to predict high-level properties, starting with a geometry that can be quickly obtained by conventional methods, rather than machine learning methods. Lu et al. [19] adopted Merck molecular force field (MMFF) [34] geometries as the starting point, to predict density functional theory (DFT) [35] properties of the molecules in the QM9 dataset. In chemical reactions, Spiekermann et al. [36] exploited reactant and product geometries to assess reaction barrier height rather than reactant and transition state (TS) geometries; because obtaining the TS geometry is computationally challenging. In addition, Chanussot et al. [23] proposed the Open Catalyst challenge. In this challenge, the IS2RE task uses initial structures (IS) for geometry optimization to predict the relaxed energies (RE) of the corresponding relaxed structures (RS). In this case, the IS and the RE can be mapped into easy-to-obtain geometries and high-level properties, respectively. Various approaches have been proposed to address this challenge [16, 27, 37].

GeoTMI shares the same goal as these previous works. However, it is important to emphasize that we propose a training framework based on a theoretical basis that possesses the capacity to be applicable to various tasks, rather than being limited to a specific task.

## 2.2 Denoising approaches in GNN

Denoising is a commonly used approach for representation learning by recovering correct data from corrupted data. Previous studies have shown that models can learn desirable representations by mapping from a corrupted data manifold to a correct data manifold. Traditional denoising auto-encoders (DAEs) employed a straightforward procedure of recovering a correct data from corrupted data thus maximizes the MI between correct data and its representation [24, 25]. Recently, several studies in GNNs have adopted denoising strategies for representation learning and robustness during training [28, 29, 38, 39, 40]. For instance, Noisy Nodes [27], which primary aim is addressing oversmoothing in GNNs, used denoising noisy node information as an auxiliary task, resulting in improved performance in property prediction. Additionally, LaGraph [41] leveraged predictive self-supervised learning of GNNs to predict the intractable latent graph that represents semantic information of an observed graph, by introducing a surrogate loss originated from image representation learning [42]. While typical denoising approaches focus on learning representations of expensive $X$, GeoTMI aims to learn higher-quality representations for $\tilde{X}$, lying on a geometrically corrupted data manifold, to predict $Y$. For this purpose, GeoTMI adopts the maximization of the three-term MI between $X$, $\tilde{X}$, and $Y$ with theoretical basis.

## 2.3 Invariant 3D GNNs for quantum chemical properties

In the field of chemistry, GNNs utilizing 3D geometric information have shown promising performance in predicting quantum chemical or systematic properties [7, 13, 14, 15, 37, 43, 44, 45]. Since target physical quantities, such as an energy, are invariant to alignments of a molecule, 3D GNN models utilize roto-translational invariant 3D information as their inputs [46, 47]. As a representative example, a distance matrix guarantees the invariance because the roto-translational transformation does not vary distances. SchNet [7, 43] and EGNN [15] are proper examples of utilizing the distance matrix. The former exploits the radial basis function based on the distance matrix, while the latter uses distance information directly on the GNN message-passing scheme. In DimeNet++ [44], along with the distance matrix, bond angles are also available as invariant 3D information. In addition, ComENet [45] and SphereNet [14] introduced dihedral angles in addition to the distance and bond angle information. Recently, several approaches such as Equiformer [16] explicitly considered irreducible representations to construct roto-translational equivariant neural networks [48, 49]. Our evaluation showed that GeoTMI is model-agnostic, hence can be easily applied to various 3D GNNs.

# 3 Method

In this section, we describe the overall framework of our proposed GeoTMI with theoretical background. First, in Section 3.1, we introduce the problem setting, i.e., the physical relationships required to predict a property from a corrupted geometry. Then, in Section 3.2, we introduce our training objective for three-term MI, which differs from the objective of typical supervised learning. Since MI itself is intractable, we derive a tractable loss for the training objective in Section 3.3. Finally, in Section 3.4, we illustrate the practical application of GeoTMI framework in the training and inference processes.

## 3.1 Problem setup

We first introduce physical relationship between our data: corrupted geometry, $\tilde{X}$, correct geometry, $X$, and quantum chemical property, $Y$. Our training data, $\mathcal{D}$, consist of observed samples $(\tilde{x}, x, y)$ from the triplet of three random variables $(\tilde{X}, X, Y) \sim q(\tilde{X}, X, Y)$. We assume that these three variables are interlinked through a Markov chain $\tilde{X} \to X \to Y$. In our problem setting, $\tilde{Z}$ and $Z$ denote representations of $\tilde{X}$ and $X$, respectively, whose probability distributions are parameterized by $\theta$, $p_\theta(\tilde{Z}, Z, Y)$.

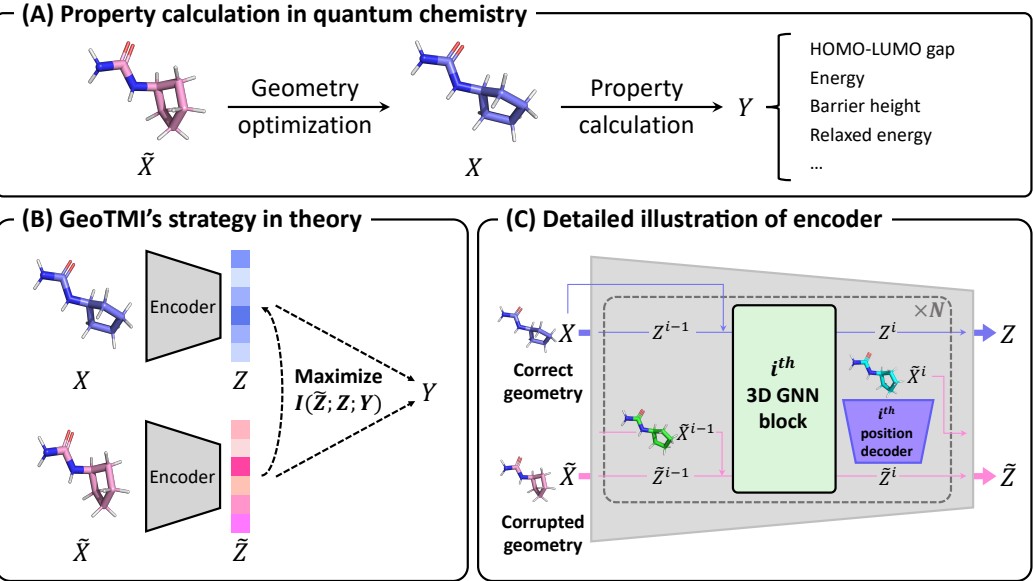

Figure 1: (A) Physical relationship between $\tilde{X}$, $X$, and $Y$ in property calculation in quantum chemistry. (B) Schematic illustration of GeoTMI's strategy in theory, where objective is maximizing three-term MI, $I(\tilde{Z}; Z; Y)$. (C) Detailed illustration of encoder architecture in practical strategy of GeoTMI. Training process employs both blue and pink lines, while inference process utilizes only pink line. All molecular geometries were plotted using PyMOL [50].

Within the standard computational chemistry process, $Y$ is obtained from the correct geometry $X$, which is acquired through geometric optimization of $\tilde{X}$ as shown in Figure 1(A). This process naturally gives rise to the Markov chain assumption, which suggests that $X$ encapsulates all the essential information for $Y$. We can establish two assumptions employing the underlying physical relationship between the variables as an inductive bias. First, there exists a higher quality of information pertaining to $Y$ within $X$ compared to $\tilde{X}$, or, more precisely, the MI between $X$ and $Y$ that is equal to or greater than the MI between $\tilde{X}$ and $Y$. This naturally follows from the property of conditional independence between non-adjacent states in a Markov chain. Second, the data distribution of $Y$ is solely dependent on $X$, irrespective of the presence of $\tilde{X}$, implying that $\tilde{X}$ and $Y$ are conditional independent given $X$.

The goal of GeoTMI is to obtain a proper representation $\tilde{Z}$ in predicting $Y$, by aligning it into $Z$ that contains more enriching information for $Y$. GeoTMI differs to self-supervised learning by emphasizing a specialized representation that is tailored to the target property. Also, acquisition of physical relationship between the variables as inductive bias leads to a higher quality representation than focusing on predicting $Y$ using $\tilde{X}$ alone.

### 3.2 Training objective

We propose a training framework for learning a proper representation for predicting $Y$ from $\tilde{X}$, which can be done by maximizing the MI between the variables, $I_\theta(\tilde{Z}; Y)$. This is somewhat similar to the objective of the general supervised learning which predicts $Y$ from $\tilde{X}$. However, the training only with $\tilde{X}$ to predict $Y$ could be erroneous because there is no guarantee a model utilize the proper information resided in both $\tilde{X}$ and $X$.

If we introduce $Z$, one can express $I_\theta(\tilde{Z}; Y)$ as following:

$$I_\theta(\tilde{Z}; Y) = I_\theta(\tilde{Z}; Y|Z) + I_\theta(\tilde{Z}; Z; Y). \tag{1}$$

In Equation (1), the conditional MI $I_\theta(\tilde{Z}; Y|Z)$ implies undesirable information of $\tilde{Z}$ in predicting $Y$ that is not relevant to $Z$. This is a direct counterpart to the physical inductive bias in the previous

section, where $X$ is sufficient information for the prediction of $Y$, and thus should be minimized to zero in the optimal case (see Appendix A.1). Maximizing $I_\theta(\tilde{Z}; Y)$ while maintaining the zero inductive bias is ideal, but non-trivial and challenging. To address this, we propose a straightforward solution by introducing the inductive bias as a regularization term, which reformulates our initial objective into maximization of the three-term MI,

$$I_\theta(\tilde{Z}; Z; Y) = I_\theta(\tilde{Z}; Y) - I_\theta(\tilde{Z}; Y|Z).$$

### 3.3 Tractable loss derivation

In general, MI is not tractable, and accurately estimating it is another challenging task. Thus, we have derived a tractable lower bound of the three-term MI, allowing us to practically maximize it (see Figure 1(B)).

**Proposition 3.1.** $I(\tilde{Z}; Z; Y) \geq \text{LB} + H(Y)$ *for any triplets of random variables* $(\tilde{Z}, Z, Y)$*, where* $\text{LB} = -H(Y|Z) - \frac{1}{2}H(Y|\tilde{Z}) - \frac{1}{2}H(Z|\tilde{Z})$.

Since $H(Y)$ is constant term in respect of model parameters, we have the three distincts optimization targets: (1) property from corrupted representation $H(Y|\tilde{Z})$, (2) property from correct representation $H(Y|Z)$, and (3) reconstruction to correct representation $H(Z|\tilde{Z})$.

The conditional entropy term related to the property is estimated by a parameterized distribution $p_{\pi_1}$ based on the positiveness of KL divergence:

$$
\begin{aligned}
-H(Y|Z) - H(Y|\tilde{Z}) &= \mathbb{E}_{p_\theta(\tilde{Z}, Z, Y)}\left[\log p_\theta(Y|\tilde{Z}) + \log p_\theta(Y|Z)\right] \\
&\geq \mathbb{E}_{p_\theta(\tilde{Z}, Z, Y)}\left[\log p_{\pi_1}(Y|\tilde{Z}) + \log p_{\pi_1}(Y|Z)\right] \\
&\sim -\sum\left(\mathcal{L}\left(y, h_{\pi_1}(\tilde{z})\right) + \mathcal{L}\left(y, h_{\pi_1}(z)\right)\right).
\end{aligned}
$$

Here, we introduce property predictor $h_{\pi_1}$ which is parameterized by $\pi_1$. Similarly, the other term is estimated by a parameterized distribution $p_{\pi_2}$,

$$
\begin{aligned}
-H(Z|\tilde{Z}) &= \mathbb{E}_{p_\theta(\tilde{Z}, Z, Y)}\left[\log p_\theta(Z|\tilde{Z})\right] \\
&\geq \mathbb{E}_{p_\theta(\tilde{Z}, Z, Y)}\left[\log p_{\pi_2}(Z|\tilde{Z})\right] \\
&\sim -\sum \mathcal{L}(z, \hat{g}_{\pi_2}(\tilde{z})).
\end{aligned}
$$

Here, $\hat{g}_{\pi_2} : \tilde{Z} \to Z$ denotes a parametric decoder for information flows. Since $Z$ is a parameterized variable which is not optimal in an initial training stage, the optimization could be unstable. If we assume the encoder $f_\theta : X \to Z$ is continuous bijective, we could introduce a surrogate loss of decoding $\tilde{Z}$ into $X$,

$$\mathbb{E}_{p_\theta(\tilde{Z}, Z, Y)}\left[\mathcal{L}\left(f_\theta^{-1}(Z), f_\theta^{-1} \circ \hat{g}_{\pi_2}(\tilde{Z})\right)\right] = \mathbb{E}_{p_\theta(\tilde{Z}, Z, Y)}\left[\mathcal{L}\left(X, g_{\pi_2}(\tilde{Z})\right)\right],$$

where $g_{\pi_2} = f_\theta^{-1} \circ \hat{g}_{\pi_2} : \tilde{Z} \to X$ denotes a decoder reconstructing $X$ rather than $Z$. It still maximizes MI between $\tilde{Z}$ and $Z$, in that the continuous and bijective mapping does not change the MI. In summary, the training process is about finding optimal model parameters $\theta$, $\pi_1$, and $\pi_2$ to minimize the following:

$$\mathbb{E}_{p_\theta(Z, \tilde{Z}, Y)}\left[\underbrace{\mathcal{L}\left(Y, h_{\pi_1}(\tilde{Z})\right)}_{\mathcal{L}_{\text{y,corrupted}}} + \underbrace{\mathcal{L}\left(Y, h_{\pi_1}(Z)\right)}_{\mathcal{L}_{\text{y,correct}}} + \underbrace{\mathcal{L}\left(X, g_{\pi_2}(\tilde{Z})\right)}_{\mathcal{L}_d}\right].$$

The tractable loss function comprises the three terms: $\mathcal{L}_{\text{y,corrupted}}$, $\mathcal{L}_{\text{y,correct}}$, and $\mathcal{L}_d$. We refer to $\mathcal{L}_\text{y}$ as the property prediction loss and $\mathcal{L}_d$ as the denoising loss. We chose the absolute error for the loss function $\mathcal{L}$. The proof of Proposition 3.1 and details of the denoising loss are described in Appendix A.

### 3.4 Overall framework

The proposing framework comprises the encoder, predictor, and decoder. The encoder maps molecular geometries to their representations, while the predictor estimates target properties, and the decoder restores the molecular geometries. The encoder design involves 3D GNN layers for both $X$ and $\tilde{X}$, sharing model parameters. It is appropriate approach because $X$ and $\tilde{X}$ belong to the same data modality. In addition, the encoder for $\tilde{X}$ includes explicit position update layers that are inspired by the geometry optimization process. The effect of intermediate geometries as input is studied in Appendix B.1. The practical model architecture including the encoder design is depicted in Figure 1(C).

In practice, an auxiliary loss is introduced as an add-on for the denoising loss to softly guide the position update toward $X$. We will refer to this as gradual denoising loss, which measures the difference between each updated geometry and the corresponding linearly interpolated target geometry. The details and ablation study of this are in Appendix B.2. We chose the same architecture of the position update layer for the decoder.

During training, $\tilde{X}$ and $X$ are mapped to $\tilde{Z}$ and $Z$ respectively. The property prediction loss is computed based on the results from $\tilde{Z}$ and $Z$, while the denoising loss involves reconstruction of $X$ from $\tilde{Z}$. In inference process, only $\tilde{Z}$ encoded by $\tilde{X}$ is used for property prediction. It is noteworthy that GeoTMI introduces a novel representation learning approach that leverages $X$ and $Z$ for robust property prediction, and its effectiveness lies in not requiring $X$ and $Z$ during the inference process.

## 4 Experiments

We have tried to demonstrate the effectiveness of GeoTMI in providing a new solution to the infeasibility of high-level 3D geometry, rather than focusing on the performance of the state-of-the-art GNN architecture itself. Thus, in this section, we have focused on showing the applicability of GeoTMI to a variety of GNN architectures and its effectiveness in predicting properties in various areas of chemistry. The tasks and architectures tested were selected based on computational cost and memory efficiency, as well as model performance. All experiments were conducted using RTX 2080 Ti GPU with 12 GB of memory, RTX 3080 Ti GPU with 12 GB of memory, or RTX A4000 GPU with 16 GB of memory. GNN models were trained on a single GPU, except for those in the IS2RE task of OC20, where we used eight RTX A4000 GPUs.

### 4.1 Molecular property prediction

Predicting molecular properties is crucial to various fields in chemistry. The QM9 [30] is widely used benchmark dataset for molecular property prediction comprised of 134k molecule information; each molecule consists of at most nine heavy atoms (C, N, O, and F). Each data sample contains optimized geometry and more than 10 corresponding DFT properties.

This study focuses on the $\text{QM9}_\text{M}$ [19] task which predicts DFT properties using the MMFF geometry. The $\text{QM9}_\text{M}$ dataset originated from the QM9 differs only in the molecular geometry; each geometry herein has been obtained with additional MMFF optimization starting with the corresponding geometry in the QM9. Here, the MMFF geometry is regarded as a relatively easy-to-obtain geometry compared to the DFT geometry.

**Training setup**. This study employed the following three GNNs using distinct 3D information to demonstrate the effectiveness of GeoTMI: EGNN [15] (implementation follows [51]), SchNet [43], and DimeNet++ [44]. We appended the position update to the SchNet and DimeNet++ to ensure that the denoising process can be applied to them in the same manner in the coordinate update of the EGNN. To train the models, we considered the DFT geometry from the QM9 dataset as $X$, and the corresponding MMFF geometry from $\text{QM9}_\text{M}$ dataset as $\tilde{X}$.

Molecular 2D graph information is similar to MMFF geometry information in that it is also more readily available than DFT geometry. There have been many attempts to predict accurate molecular properties from 2D graphs alone [17, 52, 53]. Recently, Luo et al. [17] developed the Transformer-M model, which can utilize both 2D graph and 3D geometry information in training to predict molecular properties with high accuracy using only 2D graphs. To compare the usefulness of 2D graphs

and MMFF geometries as easy-to-obtain inputs, we evaluated the prediction performance of the Transformer-M model on 2D graph inputs without pre-training. Note that the Transformer-M model reported their performance using $X$ based on pre-training in the original paper.

For all tested models, we used 100,000, 18,000, and 13,000 molecular data for training, validation, and testing, respectively, as in previous work by Satorras et al. [15]. The detailed hyperparameters of each model are introduced in Appendix C.1.

**Results on molecular property prediction**. Table 1 shows the prediction accuracy according to input types and models. Results for SchNet and DimeNet++ are shown in Appendix B.3. GeoTMI achieved performance improvements across all properties and models. For example, GeoTMI resulted in accuracy improvements of 7.0∼27.1% for EGNN, as shown in Table 1. Meanwhile, Transformer-M trained using both 2D graphs and $X$ resulted in accuracy improvements of -15∼21% compared to the same model trained using 2D graphs only. Despite the similar prediction performance of Transformer-M and EGNN based on $X$, it is noteworthy that for most properties, the Transformer-M models using 2D graphs for prediction were less accurate than the 3D GNNs tested. This result implies that while both the MMFF geometry and the molecular the 2D graph are easy-to-obtain inputs, the MMFF geometry contains more useful information for learning the relationship between molecules and their quantum chemical properties. Furthermore, we conducted additional experiments for three properties ($\mu$, $R^2$, and $U_0$) using scaffold-based splitting, a methodology that offers a more realistic and demanding setting for evaluating out-of-distribution (OOD) generalization (see Appendix B.4). Once again, GeoTMI consistently improved its prediction performance, highlighting the robustness of GeoTMI.

Table 1: MAEs for QM9's properties. The best performance among the models that do not use $X$ in the inference (Infer.) process is shown in bold. The values of Transformer-M using $X$ were borrowed from Luo et al. [17]. The performance of GeoTMI integrated with SchNet and DimeNet++ is provided in Appendix B.3.

| Methods | Input type (Train / Infer.) | $U_0$ (meV) | $\mu$ (D) | $\alpha$ (Bohr$^3$) | $\epsilon_{\text{HOMO}}$ (meV) | $\epsilon_{\text{LUMO}}$ (meV) | GAP (meV) | $R^2$ (Bohr$^2$) | $C_v$ ($\frac{cal}{mol \cdot K}$) | ZPVE (meV) |
|---|---|---|---|---|---|---|---|---|---|---|
| Transformer-M [17] | $X/X$ | 14.8 | - | - | 26.5 | 23.8 | - | - | - | - |
| EGNN | $X/X$ | 12.9 | 0.0350 | 0.0759 | 31.2 | 26.6 | 51.1 | 0.130 | 0.0336 | 1.59 |
| Transformer-M | 2D / 2D | 38.2 | 0.309 | 0.171 | 53.6 | 52.5 | 77.1 | 11.4 | 0.0669 | 4.79 |
| Transformer-M | 2D, $X$ / 2D | 43.9 | 0.245 | 0.160 | 48.7 | 46.3 | 68.4 | 10.3 | 0.0683 | 3.85 |
| EGNN | $\tilde{X}/\tilde{X}$ | 17.4 | 0.133 | 0.125 | 38.4 | 34.4 | 58.0 | 5.60 | 0.0445 | 1.97 |
| EGNN + GeoTMI | $X, \tilde{X}/\tilde{X}$ | **14.5** | **0.100** | **0.105** | **35.7** | **31.2** | **53.2** | **4.08** | **0.0407** | **1.76** |
| Improvements by GeoTMI (%) | | 16.7 | 24.8 | 16.0 | 7.03 | 9.30 | 8.28 | 27.1 | 8.54 | 10.7 |

## 4.2 Reaction property prediction

A chemical reaction is a process in which reactant molecules convert to product molecules, passing through their TSs. Predicting properties related to the reaction is important for understanding the nature of the chemistry [31]. The barrier height, as one of the reaction properties, is defined as the energy difference between the geometry of the TS, $X^{TS}$, and the geometry of the reactant, $X^R$. Commonly, optimizing a $X^{TS}$ utilizes both $X^R$ and the geometry of the product $X^P$. However, this optimization process is typically resource-intensive, requiring approximately 10 times more computational resources than optimizing $X^R$ or $X^P$ individually [54]. Thus, predicting accurate barrier height without $X^{TS}$ is necessary to reduce computational costs.

From this point of view, we focused on the task to predict DFT calculation-based barrier heights using $X^R$ and $X^P$ for the elementary reaction of the gas phase, as reported by Spiekermann et al. [36]. In contrast to most molecular properties, the property is a function of not just a single molecular geometry, but ($X^R$, $X^{TS}$), which can be interpreted as an optimized version of ($X^R$, $X^P$). Thus, we considered that $X := (X^R, X^{TS})$ and $\tilde{X} := (X^R, X^P)$ in this task.

We used two datasets, released by Grambow et al. [31], for comparison with the previous work. The first dataset consists of unimolecular reactions, namely CCSD(T)-UNI. The second dataset, B97-D3, has 16,365 reactions.

**Training setup**. Spiekermann et al. [55] proposed two models for predicting the barrier height using the 2D and 3D information of $\tilde{X}$, respectively. They used D-MPNN for 2D GNN and DimeNet++ for 3D GNN, which will be referred to as the 2D D-MPNN model and the 3D DimeReaction (DimeRxn), respectively. Here, the DimeRxn trained with $\tilde{X}$ showed lower performance than the 2D D-MPNN because the 3D GNNs were sensitive to the noise in the input geometry, as pointed out in another study [56]. Thus, our method, which removes the noise, can be useful for DimeRxn.

For the DimeRxn, we also adopted EGNN's coordinate update scheme as a decoder to predict correct geometries. We trained the D-MPNN model, and the DimeRxn models without and with GeoTMI for CCSD(T)-UNI and B97-D3 datasets. The used data split and augmentation were the same as in the previous work by Spiekermann et al. [55]. In particular, we note that scaffold splitting was used on the datasets to evaluate the OOD generalization ability of the model. The hyperparameters used are described in Appendix C.1.

Table 2: MAEs for predicted reaction barrier ($\mathrm{kcal/mol}$). The best performance among the models that do not use $X$ in the inference (Infer.) process is shown in bold.

| Methods | Input type | Dataset | |
| --- | --- | --- | --- |
| | (Train / Infer.) | CCSD(T)-UNI | B97-D3 |
| DimeRxn | $X/X$ | 2.38 | 1.92 |
| D-MPNN | 2D / 2D | 4.59 | 4.91 |
| DimeRxn | $\tilde{X}/\tilde{X}$ | 6.03 | 7.32 |
| DimeRxn + GeoTMI | $X, \tilde{X}/\tilde{X}$ | **3.90** | **4.17** |
| Improvements by GeoTMI (%) | | 35.3 | 43.0 |

**Results on reaction property prediction**. Table 2 shows the results of prediction accuracy according to input types and models. The DimeRxn trained with $X$ has the best prediction performance for all methods, while DimeRxn trained with $\tilde{X}$ has the worst prediction performance. The result supports that DimeRxn is highly dependent on the quality of input geometry, as previously mentioned. Thus, as we expected, GeoTMI, which is developed for learning a proper representation for predicting $Y$ from $\tilde{X}$, induced accuracy improvements of 35.4% and 43.0% than DimeRxn without GeoTMI in terms of MAE for the CCSD(T)-UNI and B97-D3 datasets, respectively. The results show that it outperforms the 2D D-MPNN model, again demonstrating the usefulness of the 3D easy-to-obtain geometry with GeoTMI, which is identified in the previous section.

## 4.3 IS2RE prediction

The OC20 dataset contains data consisting of the slab called a catalyst and molecules called adsorbates for each of the systems. There are more atoms and a wider variety of atom types compared to previously studied datasets. In detail, the dataset contains more than 460k pairs of IS, RS, and RE. We focus on the IS2RE task, which is to predict the RE using the IS. From the perspective of computational chemistry, the RS are obtained through costly quantum chemical calculations based on the IS. Thus, in this task, we considered IS as $\tilde{X}$ and RS as $X$.

**Training setup**. We adopted the Equiformer model [16] to evaluate the effectiveness of GeoTMI in the IS2RE task. The Equiformer model achieved state-of-the-art performance by using Noisy Nodes, where the IS2RS auxiliary task was integrated with the IS2RE task. We note that the hyperparameters used are the same as in the previous work, except for the number of transformer blocks to train each model, due to the limitation of our computational resources. We refer to the model trained only on the IS2RE task without Noisy Nodes as the baseline model, namely Equiformer*. We performed a comparative analysis of three training frameworks: (1) Equiformer*, (2) Equiformer* + Noisy Nodes, and (3) Equiformer* + GeoTMI. Thus, this evaluation with Equiformer can show the effectiveness of GeoTMI on the baseline model while allowing for comparison with Noisy Nodes.

To implement the Equiformer with GeoTMI, we followed much of the original Equiformer paper. First, we used the same Noisy Nodes data augmentation. Second, we used a similar node-level

auxiliary loss for the IS2RS task. The auxiliary loss predicts the node-level difference between target positions and noisy inputs, which corresponds to the denoising loss of $\mathcal{L}_d$. The different points of the "Equiformer* + GeoTMI" compared to the "Equiformer* + Noisy Nodes" are as follows. The noisy positions were explicitly updated by passing through GNN layers. The detailed objective here is to calculate the difference between the updated noisy positions and the linearly interpolated target positions at each GNN layer, which we refer to as the gradual denoising loss in our paper. In addition, we incorporated an auxiliary task that predicts the RE from the RS, denoted as $\mathcal{L}_{y,\text{correct}}$, which ultimately facilitates the training process of maximizing the three-term MI.

Table 3: Results on the OC20 IS2RE test set with different methods based on Equiformer architectures [16]. The Equiformer* denotes a model that reduces the number of transformer blocks from 18 to 4 while keeping other hyperparameters the same. The best performance among the Equiformer* models is shown in bold, and its improvement rate is shown in the last row.

| | Energy MAE (eV) $\downarrow$ | | | | | EwT (%) $\uparrow$ | | | | |
| Methods | ID | OOD Ads | OOD Cat | OOD Both | Average | ID | OOD Ads | OOD Cat | OOD Both | Average |
|---|---|---|---|---|---|---|---|---|---|---|
| Equiformer + Noisy Nodes [16] | 0.417 | 0.548 | 0.425 | 0.474 | 0.466 | 7.71 | 3.70 | 7.15 | 4.07 | 5.66 |
| Equiformer* | 0.515 | 0.651 | 0.531 | 0.603 | 0.575 | 4.81 | 2.50 | 4.45 | 2.86 | 3.66 |
| Equiformer* + Noisy Nodes | 0.449 | 0.606 | 0.460 | 0.540 | 0.513 | 6.47 | 3.04 | 5.83 | 3.52 | 4.72 |
| Equiformer* + GeoTMI | **0.425** | **0.583** | **0.440** | **0.521** | **0.492** | **7.60** | **3.86** | **6.97** | **4.03** | **5.62** |
| Improvement (%) | 17.6 | 10.5 | 17.1 | 13.7 | 14.4 | 58.0 | 54.4 | 56.6 | 40.9 | 53.8 |

**Results on IS2RE with Noisy Nodes**. We have summarized the IS2RE results in Table 3. To evaluate each method, the MAE of the RE prediction using IS and the energy within a threshold (EwT), the percentage in which the MAE of the predicted energy is within 0.02 eV, are used. Both Noisy Nodes and GeoTMI show performance improvements over the baseline Equiformer*, but GeoTMI achieves better performance gains across all metrics. Despite the improvements, the prediction performance with GeoTMI is still lower than the original model with 18 transformer blocks in most cases in terms of MAE. However, the prediction performance is similar to the original model for EwT and even better for OOD Ads.

## 4.4 Ablation study

GeoTMI uses a combination of $\mathcal{L}_d$, $\mathcal{L}_{y,\text{correct}}$, and the position update to improve the accuracy of predicting quantum chemical properties using $\tilde{X}$. To verify an individual contribution of each component of GeoTMI, we conducted ablation studies. Table 4 shows that all strategies are individually meaningful to reduce prediction error regardless of the properties. In this experiment, it is noteworthy that training without either $\mathcal{L}_{y,\text{correct}}$ or $\mathcal{L}_d$ is no longer maximizing the lower bound of the three-term MI. The prediction performance of these models performs worse than trained models using GeoTMI except for $C_v$. The results imply that our proposed three-term MI maximization is key in prediction performance based on $\tilde{X}$. Additionally, the table shows that position update, introduced

Table 4: Ablation study for GeoTMI. BH and PU denote reaction barrier height and position update, respectively. Prediction accuracy is compared in terms of MAE. The most degraded results are underlined.

| Dataset | Property | Unit | GeoTMI | w/o $\mathcal{L}_d$ | w/o PU | w/o $\mathcal{L}_{y,\text{correct}}$ |
|---|---|---|---|---|---|---|
| QM9 + QM9$_\text{M}$ | $U_0$ | meV | 14.5 | 21.0 | 14.2 | 15.2 |
| | $R^2$ | Bohr$^2$ | 4.08 | 6.43 | 4.12 | 4.43 |
| | $C_v$ | cal/mol $\cdot$ K | 0.0407 | 0.0503 | 0.0410 | 0.0401 |
| | $\mu$ | D | 0.0997 | 0.127 | 0.111 | 0.110 |
| CCSD(T)-UNI | BH | kcal/mol | 3.90 | 4.41 | 5.77 | 4.27 |
| B97-D3 | BH | kcal/mol | 4.17 | 4.49 | 7.19 | 4.45 |

by our intuition, is a key component for the BH prediction and helps increase overall prediction performance.

## 5 Conclusion and Limitations

In this study, we propose GeoTMI, a novel training framework designed to exploit easy-to-obtain geometry for accurate prediction of quantum chemical properties. The proposed framework is based on the Markov chain assumption and the theoretical basis that maximizes the mutual information (MI) between property, correct and corrupted geometries, mitigating the degradation in accuracy resulting from the use of the corrupted geometry. To achieve this, GeoTMI incorporates a denoising process to effectively address the inherent challenges associated with acquiring correct 3D geometry. In particular, we introduced the position update in the denoising process and gradual denoising loss to enhance the efficacy of the training process.

We have verified that GeoTMI consistently improves the prediction performance of 3D GNNs for three benchmark datasets. Nevertheless, there are several limitations in this work. First, GeoTMI addresses the inductive bias by incorporating a soft regularization approach instead of directly vanishing it to zero. Second, we could not perform an extensive optimal hyperparameter search due to a lack of computational resources. However, our consistent experimental results on various tasks showed the effectiveness and robustness of the GeoTMI. In this light, we envision that the GeoTMI becomes a new solution to solve the practical infeasibility of high-cost 3D geometry in many other chemistry fields.

## 6 Acknowledgement

This work was supported by the Korea Environmental Industry and Technology Institute (Grant No. RS202300219144), the Technology Innovation Program funded by the Ministry of Trade, Industry & Energy, MOTIE, Korea (Grant No. 20016007), and the Ministry of Science and ICT, Korea (Grant No. RS-2023-00257479).

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
