# A    Theoretical Basis

## A.1    Conditionally independence

**Proposition A.1.** *For any random variables $(\tilde{X}, X, Y)$, if $\tilde{X}$ and $Y$ are conditional independent given $X$, then $I(\tilde{X}; Y|X) = 0$.*

*Proof of Proposition A.1.* From the definition of conditional mutual information (MI), we start from the below.

$$I(\tilde{X}; Y|X) = H(\tilde{X}|X) + H(Y|X) - H(\tilde{X}, Y|X)$$

$$= \mathbb{E}_{p(\tilde{X}, X, Y)} \left[ -\log \frac{p(\tilde{X}|X)p(Y|X)}{p(\tilde{X}, Y|X)} \right]$$

$$= \mathbb{E}_{p(\tilde{X}, X, Y)} \left[ -\log \frac{p(\tilde{X}, X)p(X, Y)/p(X)^2}{p(\tilde{X}, X, Y)/p(X)} \right]$$

$$= \mathbb{E}_{p(\tilde{X}, X, Y)} \left[ -\log \frac{p(\tilde{X}, X)p(X, Y)}{p(\tilde{X}, X, Y)p(X)} \right]$$

$$= \mathbb{E}_{p(\tilde{X}, X, Y)} \left[ -\log \frac{p(X, Y)/p(X)}{p(\tilde{X}, X, Y)/p(\tilde{X}, X)} \right]$$

$$= \mathbb{E}_{p(\tilde{X}, X, Y)} \left[ -\log \frac{p(Y|X)}{p(Y|\tilde{X}, X)} \right]$$

$$= \mathbb{E}_{p(\tilde{X}, X, Y)} \left[ -\log \frac{p(Y|X)}{p(Y|\tilde{X}, X)} \right] \qquad \text{(conditional independence)}$$

$$= 0$$

$\square$

## A.2    Lower bound of three-term MI

Here, we derive the lower bound of three-term MI described at Proposition 3.1.

*Proof of Proposition 3.1.* We need to prove the following inequality first.

$$H(Z) - I(\tilde{Z}; Z) \geq I(Z; Y) - I(\tilde{Z}; Z; Y) \quad \forall \text{ random variables } \tilde{Z}, Z, Y$$

$$\text{LHS} = H(Z|\tilde{Z})$$

$$\text{RHS} = I(Z; Y) - [I(Z; Y) - I(Z; Y|\tilde{Z})]$$

$$= I(Z; Y|\tilde{Z})$$

$$= \mathbb{E}_{p(\tilde{Z}, Z, Y)} \left[ -\log \frac{p(Z|\tilde{Z})p(Y|\tilde{Z})}{p(Z, Y|\tilde{Z})} \right]$$

$$= \mathbb{E}_{p(\tilde{Z}, Z, Y)} \left[ -\log \frac{p(Z|\tilde{Z})}{p(Z|\tilde{Z}, Y)} \right]$$

$$= H(Z|\tilde{Z}) - H(Z|\tilde{Z}, Y)$$

Since conditional entropy is non-negative, LHS$\geq$RHS.

By applying the above, we derive the following two inequalities:

$$I(\tilde{Z}; Z; Y) \geq I(\tilde{Z}; Z) + I(Z; Y) - H(Z)$$

$$= -[H(Z) - I(\tilde{Z}; Z)] - [H(Y) - I(Z; Y)] + H(Y)$$

$$= -H(Z|\tilde{Z}) - H(Y|Z) + H(Y),$$

$$I(\tilde{Z}; Z; Y) \geq I(\tilde{Z}; Y) + I(Z; Y) - H(Y)$$
$$= -[H(Y) - I(\tilde{Z}; Y)] - [H(Y) - I(Z; Y)] + H(Y)$$
$$= -H(Y|\tilde{Z}) - H(Y|Z) + H(Y).$$

By adding the two inequalities, we derive a lower bound:

$$I(\tilde{Z}; Z; Y) \geq \underbrace{H(Y) - H(Y|Z) - \frac{1}{2}H(Y|\tilde{Z}) - \frac{1}{2}H(Z|\tilde{Z})}_{\text{LB}}.$$

$\square$

Though the coefficient of the lower bound is different for each term, our practical loss is calculated as follows:

$$\mathcal{L}_{\text{total}} = \mathcal{L}_{\text{y,corrupted}} + \mathcal{L}_{\text{y,correct}} + \lambda\mathcal{L}_d,$$

where $\mathcal{L}_{\text{y,corrupted}}$, $\mathcal{L}_{\text{y,correct}}$, and $\mathcal{L}_d$ correspond to $H(Y|\tilde{Z})$, $H(Y|Z)$, and $H(Z|\tilde{Z})$, respectively, and the coefficient $\lambda$ is adopted for a practical reason. The searching space of $\lambda$ is described in Appendix C.

### A.3 Choice of denoising loss

**Surrogate loss**. We decode $X$ from $\tilde{Z}$ as a surrogate task for $H(Z|\tilde{Z})$. Since an MI is invariant to continuous and bijective mappings, the surrogate loss to reconstruct $X$ can be obtained by transforming from continuous representation space to data space. Although general 3D GNNs do not satisfy this requirement, we have empirically confirmed robust results in various chemistry tasks using several GNNs.

Nevertheless, it is necessary to explain the difference between maximizing $I(\tilde{Z}; Z)$ and $I(X|\tilde{Z})$. As shown in Figure 1(A), various properties can be obtained from $X$, implying that $X$ contains information that is necessary to predict all the properties. However, $Z$ partially contains the information of $X$ in that it is a representation of $X$. For a specific property prediction task, the ideal situation would be for $Z$ to contain only the information necessary to accurately predict $Y$. In this context, mapping $\tilde{Z}$ to $X$ instead of $Z$ may cause superfluous information which is irrelevant to $Y$. But even so, it does no harm for our overall purpose of addressing the inductive bias of physical relationship.

**Geometric denoising loss**. We incorporated a geometric denoising loss as a surrogate loss in order to maximize the MI. Specifically, we aimed to maintain the equivariance of $X$ under rotation or translation of $\tilde{X}$. To achieve this, we employed the SE(3)-equivariant decoders in this study.

For the loss metric, we chose the MAE of the atom-pair distances. This choice was natural considering the importance of bond distances in molecular geometry compared to absolute atomic positions. Specifically, the denoising metric $\mathcal{L}$ is calculated based on the $(i, j)$ atom-pair distances $d_{ij}$ and $\tilde{d}_{ij}$ of $x$ and $\tilde{x}$, respectively, which is

$$\mathcal{L}(x, \tilde{x}) = \frac{1}{|\mathcal{E}|} \sum_{(i,j) \in \mathcal{E}} |d_{ij} - \tilde{d}_{ij}|. \tag{2}$$

Here, $\mathcal{E}$ denotes a set of edges in a graph of $\tilde{x}$. Thus, the practical denoising loss is calculated as follows:

$$\mathcal{L}_d = \frac{1}{|\mathcal{D}|} \sum_{(\tilde{x}, x, y) \in \mathcal{D}} \mathcal{L}(x, g(f(\tilde{x}))), \tag{3}$$

where $f$ and $g$ denote an encoder and a decoder, respectively, and $\mathcal{D}$ is the dataset. However, in case of the OC20 dataset, we computed the loss in atomic positions to align with the baseline model for comparisons.

To effectively minimize the denoising loss, we introduced an explicit position update at each layer, inspired by the geometry optimization of quantum chemical calculation. These position update layers facilitated the differentiation between the encoders of $\tilde{X}$ and $X$, despite sharing model parameters. It involved incorporating relatively small parameter additions to induce distinct mappings.

To achieve effective denoising and align the encoding process with the geometry optimization, we introduced a gradual denoising loss. This additional loss term guides the position updates at each layer to exhibit directional behavior by forcing the updated geometry to lie within the linear interpolation between $\tilde{X}$ and $X$. We compared the prediction performance according to the degree of corruption of the input geometry, and confirmed that the prediction performance improved as the geometry closer to $X$ was used (see Appendix B.1). It indirectly explains the reason for the introducing explicit position update and gradual loss. More details on the loss form and ablation studies of gradual denoising are provided in Appendix B.2.

## B Further analyses

### B.1 Utilization of interpolated geometries

In this study, we have assumed a Markov chain given that the correct geometry $X$ is optimized from the corrupted geometry $\tilde{X}$, and the quantum chemical property $Y$ is computed from $X$. Since $X$ is more adjacent to $Y$ in the Markov chain, predictions of $Y$ based on $X$ should be more accurate than those based on $\tilde{X}$. Furthermore, from a physical standpoint, the transition from $\tilde{X}$ to $X$ is a form of geometry optimization that could be perceived as a Markov chain. Building on these assumptions, the intermediate geometry during the optimization process naturally lies between $X$ and $\tilde{X}$ within the Markov chain.

In this section, we explored the possibility of using interpolated geometry $\frac{X+\tilde{X}}{2}$ as a surrogate intermediate geometry. We expected that the following two statements will be satisfied.

- Regardless of the types of properties, the baseline (trained with $\tilde{X}$) will always have a higher MAE than the ground truth (trained with $X$).

- The interpolated geometry between $\tilde{X}$ and $X$ is a less corrupted geometry than $\tilde{X}$. Thus, MAEs of the model using the interpolated geometry are placed between those of the baseline and the ground truth.

For the demonstration, we used three individual prediction models, where the training input for each model was $X$, $\frac{X+\tilde{X}}{2}$, and $\tilde{X}$, respectively. As expected, the results presented in Tables 5 and 6 consistently show that the predictive accuracy of the model decreases as the level of corruption in the geometry increases. This empirical evidence underscores the superiority of interpolated geometry over $\tilde{X}$ as an input for predicting various quantum chemical properties.

Thus, we can consider $\tilde{X} \rightarrow \frac{X+\tilde{X}}{2} \rightarrow X$ as a proxy of the geometry optimization process which is a Markov chain. Inspired by these findings, we integrated the explicit position update scheme of EGNN [15] and introduced an additional loss that exploits the interpolated geometry in the position update step following geometry optimization.

Table 5: MAEs for QM9's properties according to different inputs. Values were obtained using EGNN. $(\tilde{X}+X)/2$ denotes the interpolated geometry generated from the mean of the atom-pair distance of $\tilde{X}$ and $X$.

| Target | Unit | $\tilde{X}$ | $(\tilde{X}+X)/2$ | $X$ |
| --- | --- | --- | --- | --- |
| $U_0$ | meV | 17.4 | 14.2 | 12.9 |
| $\mu$ | D | 0.133 | 0.0807 | 0.0350 |
| $\alpha$ | Bohr$^3$ | 0.125 | 0.0947 | 0.0759 |
| $\epsilon_{\text{HOMO}}$ | meV | 38.4 | 33.4 | 31.2 |
| $\epsilon_{\text{LUMO}}$ | meV | 34.4 | 28.9 | 26.6 |
| GAP | meV | 58.0 | 53.0 | 51.1 |
| $R^2$ | Bohr$^2$ | 5.60 | 2.92 | 0.130 |
| $C_v$ | cal/mol $\cdot$ K | 0.0445 | 0.0377 | 0.0336 |
| ZPVE | meV | 1.97 | 1.74 | 1.59 |

Table 6: MAEs (kcal/mol) of predicted barrier heights according to different inputs. Values were obtained using DimeReaction. $(\tilde{X}+X)/2$ denotes the pairwise interpolated geometry generated from the mean of the atom-pair distance of $(X^R, X^P)$ and $(X^R, X^{TS})$.

| Dataset | $\tilde{X}$ | $(\tilde{X}+X)/2$ | $X$ |
|---|---|---|---|
| CCSD(T)-UNI | 6.49 | 5.30 | 2.38 |
| B97-D3 | 8.24 | 3.91 | 1.92 |

## B.2 Ablation study for denoising process

In GeoTMI, the denoising process proceeds throughout all GNN layers. The relationship between the number of GNN layers and the average absolute difference based on the atom-pair distance (D-MAE) between the correct geometry and the predicted geometry of the atomic positions at the last layer was investigated using the EGNN model (see Figure 2). We note that each EGNN was trained using only the denoising loss to confirm the denoising power alone.

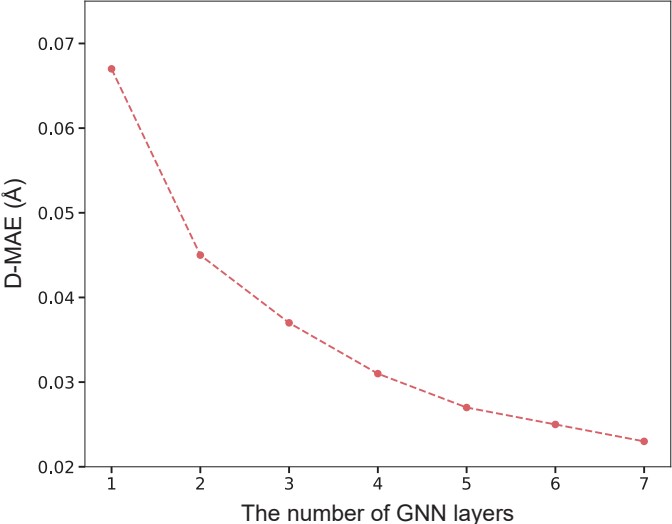

Figure 2: D-MAE according to the number of GNN layers of the EGNN. The D-MAE is measured using $X$ and the denoised $\tilde{X}$ derived from the last layer.

In particular, the denoising process with the small number of GNN layers could not restore $X$. We note that the D-MAE between the MMFF and the DFT geometries is 0.0712Å. If the denoising process couldn't restore $X$, the corresponding denoising loss can have a negative effect on learning by maintaining a large amount compared to other losses in the training process. In this respect, we designed a gradual denoising loss, which is a slightly modified version of Equation (3) with a linearly interpolated target instead of $x$. For each position update layer, a target varies linearly from $\tilde{x}$ to $x$, and the loss is calculated according to Equation (2). Specifically, the target distance of the $l$-th layer among a total of $L$ layers is as follows:

$$\bar{d}^l_{ij} = \frac{1}{L}\left(ld_{ij} + (L-l)\tilde{d}_{ij}\right).$$

Table 7 shows that the gradual denoising process is useful to increase model performance. Thus, we adopted the gradual denoising loss to predict quantum chemical properties.

## B.3 Application of GeoTMI on SchNet and DimeNet++ for QM9 and QM9$_M$ tasks

GeoTMI is applicable regardless of the 3D geometry model architectures. In this regard, we tested the effectiveness of GeoTMI using two additional 3D GNNs: SchNet [43] and DimeNet++ [44]. For

Table 7: Impact of each denoising task in terms of MAE. We tested gradual denoising ("GeoTMI"), "w/o Gradual denoising", and "Denoising last only" for four properties. When the gradual denoising was not used, the objective of all denoising processes of GNN layers is restoring $X$. The "Denoising last only" means that denoising objective contains only the denoising loss of the last GNN layer without the losses of other GNN layers.

| Property | Unit | GeoTMI | w/o Gradual denoising | Denoising last only |
|---|---|---|---|---|
| $U_0$ | meV | 14.5 | 15.4 | 14.9 |
| $R^2$ | Bohr$^2$ | 4.08 | 4.22 | 4.97 |
| $C_v$ | cal/mol · K | 0.0407 | 0.0413 | 0.0423 |
| $\mu$ | D | 0.0997 | 0.100 | 0.132 |

comparison, we also identified the performance of QM9's properties prediction from $X$ for the two models (see Table 8). The training, validation, and testing data were used as in Section 4.1.

Table 8: MAEs for QM9's properties. GeoTMI was tested with two different 3D GNNs: SchNet and DimeNet++.

| Approach | Input type (Train / Infer.) | $U_0$ (meV) | $\mu$ (D) | $\alpha$ (Bohr$^3$) | $\epsilon_{\text{HOMO}}$ (meV) | $\epsilon_{\text{LUMO}}$ (meV) | GAP (meV) | $R^2$ (Bohr$^2$) | $C_v$ ($\frac{cal}{mol \cdot K}$) | ZPVE (meV) |
|---|---|---|---|---|---|---|---|---|---|---|
| SchNet | $X/X$ | 17.0 | 0.0391 | 0.0859 | 44.3 | 34.9 | 68.6 | 0.170 | 0.0313 | 1.67 |
| DimeNet++ | $X/X$ | 8.99 | 0.0382 | 0.0583 | 25.5 | 20.8 | 43.0 | 0.342 | 0.0255 | 1.30 |
| SchNet | $\tilde{X}/\tilde{X}$ | 27.3 | 0.208 | 0.160 | 61.4 | 53.9 | 85.8 | 8.32 | 0.0595 | 2.57 |
| SchNet + GeoTMI | $X, \tilde{X}/\tilde{X}$ | 27.3 | 0.139 | 0.131 | 52.0 | 45.0 | 74.4 | 6.44 | 0.0566 | 2.38 |
| DimeNet++ | $\tilde{X}/\tilde{X}$ | 16.9 | 0.140 | 0.123 | 37.5 | 35.1 | 56.3 | 5.82 | 0.0462 | 2.10 |
| DimeNet++ + GeoTMI | $X, \tilde{X}/\tilde{X}$ | 14.0 | 0.127 | 0.109 | 35.0 | 32.6 | 55.3 | 5.29 | 0.0418 | 1.90 |
| Improvements in SchNet (%) | | 0.00 | 33.2 | 18.1 | 15.3 | 16.5 | 13.3 | 22.6 | 4.87 | 7.39 |
| Improvements in DimeNet++ (%) | | 17.2 | 9.29 | 11.4 | 6.72 | 7.12 | 1.78 | 9.11 | 9.52 | 9.52 |

## B.4 Performance of GeoTMI based on OOD data for QM9's properties

We identified the out-of-distribution (OOD) generalization ability of GeoTMI using the EGNN model on QM9's properties. To this end, we arranged 100,000, 18,000, and 13,000 molecules for training, validation, and testing, respectively, based on a scaffold split, ensuring that the molecules in the OOD were included in the test set. The split is based on the Bemis–Murcko scaffold [57] implemented in RDKit [33] library. Table 9 shows that GeoTMI has improved the model prediction performance regardless of OOD data for the tested properties. The results show the robustness of GeoTMI in terms of OOD generalization ability.

Table 9: The MAEs for $R^2$, $\mu$, and $U_0$ in the QM9$_{\text{M}}$. We verified the performance of GeoTMI on testing datasets using random and scaffold splits, respectively. For the MAE of each property, the same units are used as in Table 1.

| Split | Approach | $\mu$ | $R^2$ | $U_0$ |
|---|---|---|---|---|
| Random | EGNN | 0.133 | 5.60 | 17.4 |
| | EGNN + GeoTMI | 0.100 | 4.08 | 14.5 |
| Scaffold | EGNN | 0.195 | 10.4 | 33.0 |
| | EGNN + GeoTMI | 0.149 | 7.60 | 23.4 |

# C  Experimental details

## C.1  Parameter details

**QM9$_M$.**  We used the reported hyperparameters optimized for QM9 from previous studies for EGNN [15], SchNet [43], DimeNet++ [44], and Transformer-M [17], respectively. The search space of $\lambda$ is specified in Table 10.

Table 10: The search space of $\lambda$ on QM9$_M$ task.

| Target | EGNN | DimeNet++ | SchNet |
|---|---|---|---|
| $U_0$ | [0.1, 0.5, 1.0, 10.0] | 0.1 | [0.1, 0.5, 1.0, 5.0, 10.0] |
| $\mu$ | [0.1, 0.5, 1.0, 10.0] | 0.1 | [0.1, 0.5, 1.0, 5.0, 10.0] |
| $\alpha$ | 0.1 | 0.1 | [0.1, 0.5, 1.0, 5.0, 10.0] |
| $\epsilon_{HOMO}$ | 0.1 | 0.1 | 0.1 |
| $\epsilon_{LUMO}$ | 0.1 | 0.1 | 0.1 |
| GAP | 0.1 | 0.1 | 0.1 |
| $R^2$ | [0.1, 0.5, 1.0, 10.0] | 0.1 | [0.1, 0.5, 1.0, 5.0, 10.0] |
| $C_v$ | 0.1 | 0.1 | 0.1 |
| ZPVE | 0.1 | 0.1 | 0.1 |

**Reaction barrier prediction**.  For both DimeReaction models with and without GeoTMI, we used the same hyperparameters as Spiekermann et al. [55] except for the number of epochs and batch sizes. Table 11 shows the values of the number of epochs and batch sizes used in this work. The search space of $\lambda$ is specified in Table 12.

Table 11:  The hyperparameters used for training DimeReaction.

| Parameter | CCSD(T)-UNI | B97-D3 |
|---|---|---|
| Epoch | 200 | 200 |
| Batch size | 32 | 64 |
| Warm-up epochs | 3 | 3 |

Table 12: The search space of $\lambda$ for CCSD(T)-UNI and B97-D3 datasets.

| Parameter | CCSD(T)-UNI | B97-D3 |
|---|---|---|
| $\lambda$ | [0.005, 0.01, 0.05, 0.1, 0.5, 1.0] | [0.005, 0.01, 0.05, 0.1, 0.5, 1.0] |