# OpenReview forum: "GeoTMI: Predicting Quantum Chemical Property with Easy-to-Obtain Geometry via Positional Denoising"
_NeurIPS.cc/2023/Conference — NeurIPS 2023 poster_

### Official Review · Reviewer_rBMF · 2023-07-03

**Soundness:** 2 fair
**Presentation:** 3 good
**Contribution:** 2 fair
**Rating:** 6
**Confidence:** 4

**Summary:**

Edit: Updating score from 5 to 6 based on the discussions.

This work presents a variation of denoising autoencoder type model that uses an easy-to-obtain (corrupted) input geometry to predict properties of molecules. The assumption is that the corrupted geometry can be denoised to the correct geometry, and then used for predicting the properties of interest during training. At inference only the corrupted geometry is used, thus alleviating the need for more expensive, correct geometries. Experiments on three datasets with different choices of corrupt/correct geometries show that this approach has benefits over methods that only use the corrupt geometry for predicting properties.

**Strengths:**

* Formulating the denoising autoencoder with property prediction using a mutual information paradigm between $\tilde Z, Z, Y$ provides structure to models that use denoising AE for property prediction.

* The final objective converges to standard objectives that are commonly used in GNN literature; the method itself is easy to integrate into other GNN frameworks and as the authors claim it is _model agnostic_.

* Experiments are comprehensive on three datasets. Results show improvements compared to methods that only use the correct geometry.

**Weaknesses:**

* **Relation between $X$ and $\tilde X$**: The main assumption about the relation between $X$ and $\tilde X$ in this work is that $\tilde X$ is a _corrupted_ version of $X$, which I understand to be that $\tilde X$ contains less information than $X$. However, this empirical distinction is one of the main weaknesses of this work. What is the exact nature of relation between the two variables? Is the corruption an information destroying process?

Without a clear relation between these two variables, the complexity of predicting Y from $X$ or $\tilde X$ is difficult to appreciate. The studies in Appendix B.1 where interpolated geometries are used to predict Y from $(X+\tilde X)/2$ assumes that there is a linear relation between $X,\tilde X$ in the data space? So, when training a neural network to predict Y from $\tilde X$ via $X$ is a simpler task than being claimed in this work. Does this not simply become a supervised task  of predicting Y from $\tilde X$ with an auxiliary task of denoising to $X$ during training?

* **Model converges to denoising AE cascaded with property predictor**: Continuing with the point above, obtaining $X$ from $\tilde X$ is the classical denoising AE, and then cascading a property predictor at the output of the decoder. The authors seem to make distinctions about this setting which are unclear. The objective function in Sec. 3.3 basically converges to two property decoders (one for $\tilde Z$, one for $Z$) and a reconstruction term.

* **Selecting $\tilde X$**: As a general method, what types of $\tilde X$ and $Y$ can be used in this setting? In each of the three tasks, the selection of $\tilde X$ is based on assumptions about the complexity of these properties that are specific to each dataset. How would this generalize to other datasets/tasks?

* **Rotation and translation equivariance**: The authors state:
>> GeoTMI ensures the equivariance because it directly updates the position vector

There are no further elaborations about these claims. And how does directly updating the position vector ensure rotation and translation equivariances? Is the coordinate system also being updated? How are pairwise relations maintained when position vector of individual atoms are being updated?

**Questions:**

See Weaknesses above.

**Limitations:**

Authors have discussed limitations.

---

> ### Author Rebuttal · Authors · 2023-08-09
>
> **[Weakness 1 and 3]  “Relation between $X$, $\tilde{X}$, and $Y$” and “Selecting $\tilde{X}$”**
>
> Please refer to the second response in “Response to all reviewers”.
>
> ---
>
> **[Weakness 1 and 2] Distinction of GeoTMI from "the supervised task of predicting $Y$ from $\tilde{X}$ with an auxiliary task of denoising to $X$ during training" and "denoising AE cascaded with property predictor"**
>
> We genuinely appreciate your insightful observation and question regarding the comparison between GeoTMI and the suggested method in the context of supervised prediction from $\tilde{X}$ with an auxiliary task involving denoising to $X$.
>
> Indeed, the distinction between GeoTMI and the suggested method lies in their fundamental objectives. As a reviewer rightly pointed out, while the supervised prediction from $\tilde{X}$ through $X$ can be related to an auxiliary task of denoising, it's important to recognize that GeoTMI's core objective diverges from this.
>
> GeoTMI's objective is intuitively designed to address the specific inductive bias arising from the inherent Markov chain relationship among $\tilde{X}$, $X$, and $Y$, as highlighted in lines 127-128. This bias asserts that the information in $X$ is indispensable for predicting $Y$ when utilizing $\tilde{X}$ for predictions ($I(\tilde{X};Y\vert{X})=0$).
>
> To better predict $Y$ from $\tilde{Z}$ while preserving the corresponding Markov property in the representation space, we introduced the term $I(\tilde{Z} _\theta;Z _\theta;Y)$ as our objective (see section 3.2).
>
> The maximization of $I(\tilde{Z} _\theta;Z _\theta;Y)$ incorporates the implicit consideration of $I(\tilde{Z} _\theta;Y\vert{Z} _\theta)$, as elaborated below:
>
> $$
> \begin{aligned}
> \arg\max _\theta{I(\tilde{Z} _\theta;Z _\theta;Y)}&=\arg\max _\theta\big(I(\tilde{Z} _\theta;Y)-I(\tilde{Z} _\theta;Y\vert{Z} _\theta)\big)\newline
> &=\arg\max _\theta\big(H(Y)-H(Y\vert\tilde{Z} _\theta)-I(\tilde{Z} _\theta;Y\vert{Z} _\theta)\big) \newline
> &=\arg\max _\theta\big(-H(Y\vert\tilde{Z} _\theta)-I(\tilde{Z} _\theta;Y\vert{Z} _\theta)\big) \newline
> &=\arg\min _\theta\big(H(Y\vert\tilde{Z} _\theta)+I(\tilde{Z} _\theta;Y\vert{Z} _\theta)\big)
> \end{aligned}
> $$
>
> On the contrary, the objective of the suggested method is defined as follows, and it does not guarantee the minimization of $I(\tilde{Z} _{\theta'};Y\vert{Z} _{\theta'})$:
>
> $$
> \arg\min _{\theta'}\big({H(X\vert\tilde{Z} _{\theta'}})+H(Y\vert\tilde{Z} _{\theta'})\big)
> $$
>
> Similarly, the other suggested method, denoising AE cascaded with property predictor, also does not align with our objectives in the same reason.
>
> It's important to note that although computing the conditional mutual information $I(\tilde{Z} _\theta;Y\vert{Z} _\theta)$ is generally intractable, recent research has explored the use of neural estimators to approximate it [1]. To our best knowledge, these approaches employ maximization of a lower bound of the CMI $\hat{I} _\psi$, which translates our problem setting to a min-max problem, as depicted below. However, this approach might not be easily applicable to various tasks and models, which is why we chose the alternative objective demonstrated in section 3.3. The fact that our alternative objective might provide a looser estimation is acknowledged as one of our limitations.
>
> $$
> \begin{aligned}
> \min _{\theta}\left(H(Y\vert \tilde{Z} _\theta) + I(\tilde{Z} _\theta;Y\vert Z _\theta)\right)\ge\min _{\theta}\left(H(Y\vert \tilde{Z} _\theta) + \max _{\psi} \hat{I} _\psi\right)
> \end{aligned}
> $$
>
> We hope this provides a clearer understanding of how GeoTMI's objective addresses the specific challenges in capturing the shared information among $\tilde{Z} _\theta$*, $Z _\theta$*, and $Y$, distinct from the suggested methods.
>
> [1] S. Molavipour, G. Bassi, and M. Skoglund, in ICASSP 2020-2020 IEEE International Conference on Acoustics, Speech and Signal Processing (ICASSP) (IEEE, 2020).
>
> ---
>
> **[Weakness 4] Rotation and translation equivariance**
>
> First, we apologize for any confusion caused by the statement "GeoTMI ensures the equivariance because it directly updates the position vector ".  Thanks to your valuable feedback, we have become aware of a deficiency of detailed descriptions. The intended meaning of the statement is that GeoTMI employs well-established position vector update methods ensuring both rotation and translation equivariance.
>
> In this study, we basically used the position vector update method suggested by Satorras et al [1]. Positional vector update methods used for each task were described on lines 218, 271, and 295. The rotation and translation equivariance of each position update method is proved in "Appendix A. Equivariance Proof" in the previous work [1]. In brief, one can use the method to update the 3D Cartesian coordinates of each atom using an update vector, which is decomposed into magnitude and direction. The former is invariant, while the latter is equivariant with respect to rotation and translation of the molecular geometry. Thus, the method guarantees the equivariance of the updated molecular geometry.
>
> [1] V. G. Satorras, E. Hoogeboom, and M. Welling, in *International Conference on Machine Learning* (PMLR, 2021), pp. 9323-9332.

---

> > ### Comment · Reviewer_rBMF · 2023-08-16
> > **Response to author rebuttal**
> >
> > I thank the authors for clarifications in their rebuttal. One minor concern/clarification still remains:
> >
> > While I now see that the Markov assumption made in this work is different from a denoising A + predictor, I would like the authors to speculate about the generalization of this assumption to other tasks. Are there scenarios/applications where the relation between $\tilde{X},X,Y$ cannot be reduced to a Markov assumption? In those scenarios, how would the Geo TMI model work?

---

> > > ### Author Response · Authors · 2023-08-17
> > > **Response to reviewer's comment**
> > >
> > > We sincerely appreciate the thoughtful question offered by the reviewer. Before addressing the additional question, we would first like to reemphasize that our primary goal was to improve the prediction of high-level quantum chemical properties from easy-to-obtain geometries. To the best of our knowledge, in the field of quantum chemistry, the calculation process for certain properties typically involves geometry optimization. Consequently, the Markovian assumption can be generally consistent with the inherent nature of the problem.
> > >
> > > In scenarios where the relationship between data is non-Markovian, we could hypothetically consider both $\tilde{X}$ and $X$ as independent data $X_1$ and $X_2$ associated with property $Y$. Non-Markovianity implies that, given their respective counterparts, $X_1$ or $X_2$ is conditionally dependent on $Y$. This violates our Proposition A.1, leading to $I(X_1;Y\vert X_2)\ne0$, and also prohibits meaningful comparison between conditional entropies $H(Y\vert X_1)$ and $H(Y\vert X_2)$. Essentially, this means that predicting $Y$ necessitates information from both $X_1$ and $X_2$, not just one.
> > >
> > > Although we have not considered non-Markovian cases within the processes of obtaining quantum chemical properties in our manuscript, we have additionally thought about the several scenarios that cannot be reduced to a Markov assumption, as suggested by the reviewer. For example, in reaction barrier height problems, we can redefine $X_1$ and $X_2$ as $X^R$ and $X^{TS}$ instead of $(X^R,X^P)$ and $(X^R,X^{TS})$, respectively. The reaction barrier height fundamentally depends on both $X^R$ and $X^{TS}$, which simply leads to the definition of non-Markovian $\left(p(\mathrm{BH}\vert{X^R})\ne p(\mathrm{BH}\vert{X^R},X^{TS})\right)$. Likewise, in protein-ligand systems, predicting $\Delta\Delta{G}$ by perturbing the ligand type while keeping the same protein is another example [1]. If we denote $X_1$ and $X_2$ as complexes with different ligands interacting with the same protein, then the $\Delta\Delta{G}$ between these complexes also depends on both $X_1$ and $X_2$, reflecting non-Markovianity.
> > >
> > > Conversely to the core goal of GeoTMI (implicitly modeling the latent space to satisfy $I(X_1;Y\vert{X}_2)=0$), the aforementioned scenarios involve situations where $X_1$ and $X_2$ possess unique information that is necessary for predicting $Y$, respectively. Consequently, while it may be possible to apply GeoTMI in non-Markovian settings, it does not theoretically guarantee optimal performance due to the distinct nature of the relationship between $X_1$ and $X_2$ from our original problem setting. It is also noteworthy that the scenario of predicting $\Delta\Delta{G}$ is not related to easy-to-obtain geometry, implying that, unlike quantum chemical property prediction, the scenarios with non-Markovianity don't have to involve easy-to-obtain geometry.
> > >
> > > [1] Wang, L., Wu, Y., Deng, Y., Kim, B., Pierce, L., Krilov, G., ... & Abel, R. (2015). Accurate and reliable prediction of relative ligand binding potency in prospective drug discovery by way of a modern free-energy calculation protocol and force field. Journal of the American Chemical Society, 137(7), 2695-2703.

---

> > > > ### Comment · Reviewer_rBMF · 2023-08-18
> > > > **Response to author response**
> > > >
> > > > Thanks for the clarification. I think this sufficiently addresses my concern.
> > > >
> > > > While it is not a limitation per se, having the Markov assumption narrows the scenarios where the method is useful. This should be clarified in the manuscript. I am willing to raise my score from 5 to 6.

---

> > > > > ### Author Response · Authors · 2023-08-18
> > > > > **Response to reviewer's comment**
> > > > >
> > > > > Thank you for your understanding and feedback.
> > > > >
> > > > > We acknowledge your observation that the Markov assumption's applicability is constrained, which in turn affects the method's versatility. We will certainly incorporate this clarification into the manuscript to accurately convey the scope of the approach.

---

### Official Review · Reviewer_MDdx · 2023-07-06

**Soundness:** 4 excellent
**Presentation:** 3 good
**Contribution:** 4 excellent
**Rating:** 6
**Confidence:** 4

**Summary:**

This paper proposes a novel training framework called GeoTMI. This framework uses a denoising process to accurately predict quantum chemical properties for molecules using MMFF geometries that are much easier to obtain than DFT-optimized geometries.

**Strengths:**

1. The proposed method is interesting, and the derivation well supports the training objective.

2. The experimental results show that GeoTMI achieves very good performance over multiple molecular tasks. Although expensive DFT-optimized geometries always produce the best property prediction as expected, GeoTMI can use easy-to-obtain geometries to improve the prediction when DFT-geometries are lacking. It’s practically meaningful.


**Weaknesses:**

Although Table 3 has shown that “Equiformer + Noisy Nodes + GeoTMI” achieves better performance than “Equiformer + Noisy Nodes”, the direct comparison between Noisy Nodes and GeoTMI is missing. It would be better if a direct comparison with other denoising-based methods is included.

**Questions:**

PCQM4Mv2[1] is a quantum chemistry dataset in which DFT geometries are provided for the training set but not provided for validation and testing sets. This is a dataset/task very suitable to apply the proposed GeoTMI. It would be good if experiments on this dataset is also included.

[1]. Hu, Weihua, et al. "Ogb-lsc: A large-scale challenge for machine learning on graphs." arXiv preprint arXiv:2103.09430 (2021).

---

> ### Author Rebuttal · Authors · 2023-08-09
>
> **[Weakness 1] Although Table 3 has shown that “Equiformer + Noisy Nodes + GeoTMI” achieves better performance than “Equiformer + Noisy Nodes”, the direct comparison between Noisy Nodes and GeoTMI is missing. It would be better if a direct comparison with other denoising-based methods is included.**
>
> First, we would like to inform you that we have revised Table 3 by replacing “Equiformer + Noisy Nodes + GeoTMI” with “Equiformer + GeoTMI”. Please refer to the first response in “Response to all reviewers”.
>
> We sincerely appreciate suggestions to include a comparison with another denoising model to strengthen the evaluation of GeoTMI's performance. Unfortunately, to the best of our knowledge, all denoising works in the OC20 task have used the same node-level auxiliary loss as proposed by Noisy Nodes. We are open to any further questions or suggestions you may have.
>
> ---
>
> **[Question 1]** **PCQM4Mv2 is a quantum chemistry dataset in which DFT geometries are provided for the training set but not provided for validation and testing sets. This is a dataset/task very suitable to apply the proposed GeoTMI. It would be good if experiments on this dataset is also included.**
>
> Thank you for the suggestion regarding the PCQM4Mv2 dataset. We agree that including experiments on this dataset would indeed provide valuable insights into the performance and effectiveness of the proposed GeoTMI method. However, the main goal of the PCQM4Mv2 dataset is to achieve accurate predictions in situations where 3D DFT geometry is not available. To solve this problem, many studies have designed various model architectures to use 2D graphs as input and report the optimal hyperparameters. Since GeoTMI is designed based on 3D GNNs, we could not find a suitable model and its hyperparameters for the PCQM4Mv2 dataset, unlike our previously reported experiments (QM9, Barrier Heights, OC20). In this light, we conducted additional experiments using SchNet and its reported hyperparameters [1] to verify the effectiveness of GeoTMI on the PCQM4Mv2 dataset.
>
> The validation set provided by PCQM4Mv2 was used as the test set of our experiment. The training set provided by PCQM4Mv2 was randomly split at 6:1 to build our defined training/validation sets. We used the MMFF-optimized geometries as $\tilde{X}$. As a result, we achieved a performance improvement of 7.0973% on our test set (no DFT geometry) when using GeoTMI. We expect to see even better performance improvements on the PCQM4Mv2 dataset if we set the appropriate model and hyperparameters for optimal performance.
>
> |  Methods | Input type (Train / Infer.) | GAP (eV) |
> | --- | --- | --- |
> | SchNet | $\tilde{X}$/$\tilde{X}$ | 0.1254 |
> | SchNet + GeoTMI | $X$,$\tilde{X}$/$\tilde{X}$  | 0.1165 |
> | Improvements by GeoTMI (%) |  | 7.0973 |
>
> [1] K. T. Schütt, P. Kessel, M. Gastegger, K. A. Nicoli, A. Tkatchenko, and K. R. Muller, *J. Chem. Theory Comput.* **15**(1), 448-455 (2018).

---

> > ### Comment · Reviewer_MDdx · 2023-08-20
> >
> > Thanks for authors' response and I'd like to maintain my score.

---

### Official Review · Reviewer_PXHR · 2023-07-06

**Soundness:** 3 good
**Presentation:** 3 good
**Contribution:** 2 fair
**Rating:** 5
**Confidence:** 2

**Summary:**

The authors propose a novel method to help solve the problem of 3D positional noise in quantum chemical properties. The proposed method is like a plug-in for other 3D GNN methods to improve their performance on defective 3D positional data. The numerical results show that the model can help the GNN models to perform better on corrupted data.

**Strengths:**

1. The authors propose a novel training framework based on maximizing the mutual information between correct and corrupted geometries to make accurate predictions on noisy positional information. Involving mutual information in this problem is novel and interesting.
2. This paper has rich experimental results. From the numerical experiments, we can see that the proposed model can improve the basic GNN models on the property prediction tasks.

**Weaknesses:**

1. The main concern is that this proposed GeoMTI will be less effective with more powerful basic models. From QM9, which I'm more familiar with, GeoMTI performs best on SchNet, then EGNN, and worst on DimeNet++(an average of less than 10% improvement). Since there are more powerful molecular property prediction models such as [1], [2], [3], etc. I wonder whether the GeoMTI will be still useful on these models.
2. The motivation of this paper is a little strange. Molecules are not like the other graph problems that can easily encounter with OOD detection. Using DFT and MMFF to obtain the geometry information might lead to different molecular configurations and different energy. I'm not sure whether using MMFF to predict the DFT properties is admissible. Moreover, the DFT geometry information is not that hard to obtain, so I'm worried that the proposed problem might be a carefully tailored problem by computer scientists with no meaning to the quantum chemistry society.

[1] Spherical message passing for 3D graph networks
[2] ComENet: Towards Complete and Efficient Message Passing for 3D Molecular Graphs
[3] ViSNet: an equivariant geometry-enhanced graph neural network with vector-scalar interactive message passing for molecules

**Questions:**

1. I wonder what the performance will be if the authors give the basic prediction models the same input type as GeoMTI, which is $X$ and $\tilde{X}$ as training and $\tilde{X}$ as infer. By giving the basic models more input data, even if they might be conflicting, I think this will help the performance of these models on the corrupted data.

I might be misunderstanding something about the motivation. So further discussion is welcomed and I'm positive about changing the scores if I'm proved to be wrong.

**Limitations:**

The authors do address three limitations in the paper. However, I think the most important limitation is the motivation and usefulness of this paper.

---

> ### Author Rebuttal · Authors · 2023-08-09
>
> **[Weakness 1] From QM9 results, I wonder whether the GeoTMI will be still useful in more powerful molecular property prediction models.**
>
> We appreciate your concern about its effectiveness on more powerful models. It is important to assess GeoTMI's performance on state-of-the-art models to better understand its capabilities.
>
> Before exploring GeoTMI's performance on other GNNs, we would like to highlight two important points. First, while this evaluation was conducted on the OC20 task, we tested GeoTMI on the Equiformer, which is known as a more powerful molecular property prediction model than SchNet, EGNN, and DimeNet++ for the QM9 task. We found a significant improvement when GeoTMI was used in conjunction with Equiformer.
>
> Second, we acknowledge the relatively lower improvement observed in DimeNet++. However, this may be due to the limited hyperparameter tuning. As shown in Table 10 in the supplementary material, the search space for the hyperparameter λ was smaller in DimeNet++ compared to SchNet and EGNN. We additionally tested DimeNet++'s performance on $R^2$ and GAP with λ=1.0, which resulted in improvements of 20.4% and 5.68%, with MAEs of 4.63 $\mathrm{Bohr}^2$ and 53.1 meV, respectively. We believe these results point to potential improvements through optimal λ search.
>
> ---
>
> **[Weakness 2] The motivation is a little strange.**
>
> We would like to emphasize that our work is motivated by the need to address the high computational cost of obtaining geometry in quantum calculations. For example, in previous approaches to the QM9 task, 3D GNNs are trained to predict DFT properties from DFT geometry. However, in real-world applications, the DFT geometry is not readily available as input to 3D GNNs. If the input geometry has been obtained by DFT-based geometry optimization, the DFT properties usually already exist or can be obtained very easily. In this respect, the goal of this study is to point out the lack of practicality of 3D GNNs and to improve them. Our solution is to use geometries that can be obtained relatively cheaply, e.g. MMFF-based geometries in the QM9 task. It is important to note, however, that it is not limited to the relationship between DFT and MMFF, and is broadly applicable to similar tasks given $\tilde{X}$, $X$, and $Y$ (see the second response in "Response to all reviewers").
>
> Regarding the comment “DFT geometry information is not that hard to obtain”, we would like to tell you the following. Within DFT, there is a wide range of theory levels with different computational costs. For example, widely-used hybrid functionals like B3LYP have a time complexity of $\mathcal{O}(n^4)$. Optimization using the hybrid functionals can be time-consuming, e.g. taking over 150 days for obtaining optimized geometries on the Molecule3D test set [1]. Furthermore, the cost increases dramatically with the size of the system.
>
> Regarding the comment “whether using MMFF to predict the DFT properties is admissible”, we would like to say the following. DFT-based geometry optimization is followed by the determination of DFT properties. As a practical approach, MMFF-based optimization is often used as a preliminary step to get a good starting point to mitigate the high cost of DFT-based optimization [2-4]. Given this process, it seems admissible to use ML to directly predict DFT properties from MMFF geometries. It is also worth noting that there is extensive ML research in the quantum chemistry society to predict high-level quantum chemical properties from low-level computational results [5-9]. Moreover, as mentioned in "Related Works" section, Lu et al. [5] made the same attempt as us to predict the QM9 property from MMFF geometry.
>
> Our goal is to contribute to the advancement of both the computational chemistry and ML communities. We hope our explanation clarifies the motivation and significance of our work. We appreciate your thoughtful review and are open to any further questions or suggestions you may have.
>
> [1] Z. Xu, et al., arXiv preprint arXiv:2110.01717 (2021).
>
> [2] M. Nakata and T. Shimazaki, *J. Chem. Inf. Model.* **57**(6), 1300-1308 (2017).
>
> [3] C. A. Grambow, L. Pattanaik, and W. H. Green, *Sci. Data* **7**(1), 137 (2020).
>
> [4] S. Axelrod and R. Gomez-Bombarelli, *Sci. Data* **9**(1), 185 (2022).
>
> [5] J. Lu, C. Wang, and Y. Zhang, *J. Chem. Theory Comput.* **15**(7), 4113-4121 (2019).
>
> [6] X. García-Andrade, P. García Tahoces, J. Pérez-Ríos, and E. Martínez Núñez, *J. Phys. Chem. A* **127**(10), 2274-2283 (2023).
>
> [7] B. Savoie, Q. Zhao, D. Anstine, and O. Isayev, *Chem. Sci.* (2023).
>
> [8] R. Ramakrishnan, P. O. Dral, M. Rupp, and O. A. Von Lilienfeld, *J. Chem. Theory Comput.* **11**(5), 2087-2096 (2015).
>
> [9] K. Atz, C. Isert, M. N. Böcker, J. Jiménez-Luna, and G. Schneider, *Phys. Chem. Chem. Phys.* **24**(18), 10775-10783 (2022).
>
> ---
>
> **[Question 1] I wonder what the performance will be if the authors give the basic prediction models the same input type as GeoTMI, which is $X$ and $\tilde{X}$  as training and $\tilde{X}$ as infer.**
>
> Thanks for the suggestion to validate the effectiveness of GeoTMI. We have tried the proposed method for verifying the effectiveness in the development step. However, the accuracy improvement by the method using the EGNN for $U_0$ was -1.7 % (Improvement by GeoTMI: 16.7 %). This can be explained by the fact that multi-task learning tends to be less effective when certain tasks hold greater importance or relevance than others (predicting $Y$ from $X$ is much easier than predicting $Y$ from $\tilde{X}$). Also, the training objective is no longer maximizing the lower bound of the three-term mutual information $I(\tilde{Z}; Z; Y)$. Thus, the model based on the method can not learn the proper representation $\tilde{Z}$ in predicting $Y$, by aligning it into $Z$ that contains more enriching information for $Y$.

---

> > ### Comment · Reviewer_PXHR · 2023-08-18
> >
> > Thanks for clarifying the concerns and it seems that I have misunderstood the motivation of this paper. I will change the score accordingly.

---

### Official Review · Reviewer_WWJn · 2023-07-06

**Soundness:** 3 good
**Presentation:** 3 good
**Contribution:** 3 good
**Rating:** 6
**Confidence:** 4

**Summary:**

The paper proposes an effective framework, GeoTMI, to train 3D GNNs for quantum property prediction. Specifically, GeoTMI involves the denoising process during the learning of property prediction tasks by maximizing a three-term mutual information among the noisy representation, original representation, and prediction target. The framework effectively improves the inference performance of 3D GNNs when only easy-to-obtain (corrupted) geometry is provided during inference.

**Strengths:**

+ Comprehensive discussion with related work.

+ The idea is interesting. The major bottleneck of quantum-related tasks is the gap between the low-cost geometry and precise geometry computed by DFT algorithms. It seems that this work is pretty effective in bridging the gap.

+ The proposed approach is theoretically grounded and can provide insight for general geometry representation learning.

+ Multiple GNN models suggest generalizable effectiveness.

**Weaknesses:**

- Statement that denoising works focusing on prediction from X is not very true. It only applies to augmentation-based, but not denoising auto-encoders. The approaches also aim to perform downstream tasks given noisy/corrupted data.

- Although the work eliminates the requirement of X during inference, the precise geometry from the same dataset is still needed during training. This somehow limits the applicable scenarios and hence the impact. The authors may want to also validate the denoising effect across dataset and tasks, since the geometry is common. For example, is it possible to use a more general task (i.e., Y) and (X, \tilde{X}) pairs to pre-train the encoder, and fine-tune it on a different task but without X? If that is the case, the pre-trained encoder can be used as a foundation model with a much higher impact.

- The experimental results suggest the effectiveness of GeoTMI. However, since you mention noisy node, it may be more convincing to also compare “model+GeoTMI” with “model+noisy node”. To my knowledge, noisy node is also model agnostic.

- Could the author include some theoretical comparisons on the connection/differences between GeoTMI and related approaches to further demonstrate its supremacy? For example, would GeoTMI be any tighter bound to the three-term MI than other frameworks?

**Questions:**

* The two assumptions in the problem setup intuitively make sense to me, but could you clarify the term “higher quality of information”? Does it suggest I(X, Y)>=I(\tilde{X}, Y)?

* Could you clarify how you incorporate noisy nodes with GeoTMI? Is it a simply additive term in the loss or there is additional insights?

---

> ### Author Rebuttal · Authors · 2023-08-08
>
> **[Weakness 1] Statement that denoising works focusing on prediction from X is not very true.**
>
> We acknowledge the point that not all denoising approaches may exclusively focus on predicting $X$. However, in the context of predicting quantum chemical properties, we intended to highlight the prevalent use of denoising approaches to enhance prediction accuracy from $X$. If the reviewer could provide examples of the related studies, it would greatly assist us in refining our manuscript for better clarity.
>
> ---
>
> **[Weakness 2] Necessity of the precise geometry in training**
>
> From the perspective of computational chemistry, the quantum chemical property $Y$ is solely obtained from the molecular geometry $X$ ($Y$ can not be obtained without $X$). Thus, our goal is to develop a training framework that can make accurate $Y$ prediction from relatively easy-to-obtain molecular geometry, $\tilde{X}$, using $X$ in the training phase, but without $X$ in the inference phase.
>
> We appreciate the reviewer's concern regarding the requirement of precise geometry during training and its potential impact on applicability. Exploring ways to mitigate this limitation would be highly beneficial. Thus, we verified the usefulness of an encoder pre-trained by GeoTMI. To investigate the feasibility of GeoTMI as a pre-training method, we conducted experiments on the QM9 dataset by splitting it into two halves.
>
> For the pre-training task, the EGNN model was trained by GeoTMI and auxiliary losses for all properties of Table 1 except $\alpha$. In this case, the training objective is $I(Z;\tilde{Z}; Y)$ where $Y$ is a vector of the properties. For the fine-tuning task, the last layer of the pre-trained model was modified to predict the $\alpha$ property, and the task was performed without $X$. The same hyperparameters introduced in Appendix C.1 were used for pre-training. For comparison, we also trained the model to predict only $\alpha$ from $\tilde{X}$ without any pre-training task, and the accuracy of the model in terms of MAE was 0.178 $\mathrm{Bohr}^3$. The accuracy of the model with pre-training was 0.141 $\mathrm{Bohr}^3$. Consequently, the accuracy improvement by the pre-trained encoder was 20.7%, showing the potential of GeoTMI on a different task but without $X$.
>
> ---
>
> **[Weaknees 3], [Question 2] Relationship between Noisy nodes with GeoTMI?**
>
> We apologize for any confusion caused by the phrase "Equiformer + Noisy Nodes + GeoTMI. Please refer to the first response in “Response to all reviewers”.
>
> ---
>
> **[Weakness 4] Theoretical comparisons between GeoTMI and related approaches**
>
> To our best knowledge, no existing works have focused on maximizing three-term mutual information (MI).
>
> In our problem setting, we need to reduce the conditional mutual information $I(\tilde{Z};Y\vert{Z})$ (CMI) to satisfy the Markov property, originates from the data relationship. Recent research has concentrated on estimating two-term MI, utilizing neural network-parameterized lower bounds for estimation [1]. In particular, Molavipour et al. [2] focused on the estimation of CMI through tight lower bounds based on the Donsker-Varadhan theorem and the universal approximation theorem. We could have employed a similar CMI estimator to fulfill our primary goal as outlined in Equation 1 of our manuscript - precise estimation of conditional MI. However, this approach may be less practical for CMI minimization due to the intricacies of solving a minimax problem. Alternatively, our proposed objective provides a lower bound for the three-term MI. Although strict tightness is not guaranteed, it offers the advantage of circumventing the need for a complex minimax training process.
>
> [1] M.I. Belghazi et al., in International Conference on Machine Learning (PMLR, 2018).
>
> [2] S. Molavipour, G. Bassi, and M. Skoglund, in ICASSP 2020-2020 IEEE International Conference on Acoustics, Speech and Signal Processing (ICASSP) (IEEE, 2020).
>
> ---
>
> **[Question 1] The meaning of “higher quality of information”**
>
> As the reviewer suggests, the term *"higher quality of information"* implies that $I(X;Y) \ge I(\tilde{X};Y)$. To see detailed descriptions, please refer to the second response in “Response to all reviewers”.

---

### Author Rebuttal · Authors · 2023-08-09

$\Large{\text{Response to all reviewers}}$

We extend our sincere appreciation to the reviewers for your invaluable insights and constructive feedback, which have significantly enhanced the quality and rigor of our manuscript.  Your feedback will undoubtedly contribute to the refinement of our research. We here reply two shared questions from several reviewers.

**1. Response to questions about comparison with Noisy Nodes in Table 3**

First of all, we would like to apologize for any confusion caused by the phrase "Equiformer + Noisy Nodes + GeoTMI”. We note that the Noisy Nodes used data augmentation in the OC20 task by using multiple geometries, interpolated between the initial structure (IS) and the relaxed structure (RS), as $\tilde{X}$. To ensure a fair comparison with Noisy Nodes, we also used the same data augmentation technique. In this context, we expressed our results as "Equiformer + Noisy Nodes + GeoTMI" in the previous manuscript.

To implement the Equiformer with GeoTMI, we borrowed a lot from the original Equiformer paper [1]. First, we used the same Noisy Nodes data augmentation. Second, we used a similar node-level auxiliary loss for the IS2RS task. The auxiliary loss predicts the node-level difference between target positions and noisy inputs, which corresponds to the denoising loss of $\mathcal{L}_d$ in our notation.

The different points of the Equiformer with GeoTMI compared to the Equiformer with Noisy Nodes are as follows. The noisy positions were explicitly updated by passing through GNN layers. The detailed objective here is to calculate the difference between the updated noisy positions and the linearly interpolated target positions at each GNN layer, which we refer to as the gradual denoising loss in our paper. In addition, we incorporated an auxiliary task that predicts the relaxed energy (RE) from the relaxed structure (RS), denoted as $\mathcal{L}_{\text{y, correct}}$, which ultimately facilitates the training process of maximizing the three-term mutual information (TMI).

In this regard, after careful consideration, we realize that the phrase "Equiformer + Noisy Nodes + GeoTMI" may be misleading. While we adopted the data augmentation technique from Noisy Nodes, it is important to note that the key feature of GeoTMI is the use of a TMI approach. For clarity, we will revise Table 3 and the descriptions to make the following correction: the correct presentation of our results should be "Equiformer + GeoTMI", with the explicit clarification that we used the same data augmentation as in Noisy Nodes. We hope the revision will provide readers with more convincing and clear insights.

[1] Y. L. Liao and T. Smidt, arXiv preprint arXiv:2206.11990 (2022)

---
**2. Response to questions about the theoretical background of GeoTMI**

As stated in lines 127-128, we consider the relation of $\tilde{X}\to{X}\to{Y}$ as a Markov chain based on physical relationships. By utilizing the property of conditional independence between two non-adjacent states in a Markov chain, we derive the relationship between conditional entropy as follows:

$\begin{aligned}H(Y\vert\tilde{X}) &=\mathbb{E}_{p(\tilde{X},Y)}\big[\log{p(Y\vert\tilde{X})}\big]\newline&=\mathbb{E} _{p(\tilde{X},X,Y)}\big[\log{p(Y\vert\tilde{X})}\big]\newline&=\mathbb{E} _{p(\tilde{X},X,Y)}\big[\log{p(Y\vert{X})\big]+\mathbb{E} _{p(\tilde{X},X,Y)}\big[\log{p(X\vert\tilde{X})}}\big]\newline&=\mathbb{E} _{p(X,Y)}\big[\log{p(Y\vert{X})\big]+\mathbb{E} _{p(\tilde{X},X)}\big[\log{p(X\vert\tilde{X})}}\big]\newline&=H(Y\vert{X})+H(X\vert\tilde{X})\newline&\ge H(Y\vert X)\end{aligned}$

Intuitively, $H(Y\vert{X})$ is smaller than $H(Y\vert\tilde{X})$ since $Y$ is a property calculated from $X$. Thus, the term *"higher quality of information"* implies that $I(X;Y) \ge I(\tilde{X};Y)$, which is equivalent to $H(Y\vert\tilde{X})\ge H(Y\vert X)$. Continuing with the context, we can consider $\tilde{X}\to{\frac{X+\tilde{X}}{2}}\to{X}$ as a geometry optimization process which is a Markov chain. Appendix B.1 illustrates the uncertainty in predicting $Y$ from the state $\frac{X+\tilde{X}}{2}$, which is later compared to the state $\tilde{X}$ in a multi-state Markov chain. Thus, the later state exhibits reduced uncertainty in predicting $Y$, showing consistency with the results presented in Appendix B.1. Therefore, based on the formulation and results, we can conclude that $\tilde{X}$ should be selected as the previous state of $X$ in a Markov chain $\tilde{X}\to{X}\to{Y}$. In all of our experiments, the initial geometries $\tilde{X}$ are used for the geometry optimization to obtain optimized geometries $X$, and properties $Y$ are directly computed from $X$.

| EXP | $\tilde{X}$ | $X$ |
| --- | --- | --- |
| $\mathrm{QM9}_M$ | MMFF-optimized geometries | DFT-optimized geometries |
| Reaction | Reactant and product geometries | Reactant and transition state geometries |
| OC20 | Initial structures (IS) | Relaxed structures (RS) |

---

> ### Comment · Area_Chair_3k2R · 2023-08-13
> **lets followup for author response**
>
> Hi all,
>
> Thanks for serving as the reviewers for this submission. As the authors have already provided their responses. Now let's start further discussion. Here is a to-do list:
>
> (1) Please acknowledge the authors when you finish reading their responses.
> (2) Please indicate whether you have any further questions for the authors such that they can continue to response.
> (3) Please indicate whether you are willing to change the ratings.
>
> best,
> The AC

---

### Decision · Program_Chairs · 2023-09-21

**Decision:**

Accept (poster)

**Comment:**

This paper proposes GeoMTI to help learn the 3D graph embedding from easy-to-obtain (corrupted) data to predict the properties of molecules. Experiments on three datasets show that the proposed method can help to provide accurate prediction only with corrupted data during inference. The authors propose a novel and interesting method, which uses mutual information to build up a connection between the correct data and the corrupted data. The proposed method bridges the gap between low-cost geometry and precise prediction, which can also provide insights into the general geometry representation learning in quantum chemistry. Experiment results on three datasets with different combinations of training and test sets illustrate the capability of the proposed model.

Several problems regarding the performance with a better basis GNN model and comparison with other related approaches are raised by the reviewers and are solved by the authors' rebuttal.